# Isolation of a New Infectious Pancreatic Necrosis Virus (IPNV) Variant from Genetically Resistant Farmed Atlantic Salmon (*Salmo salar*) during 2021–2022

**DOI:** 10.3390/pathogens11111368

**Published:** 2022-11-16

**Authors:** Marcos Godoy, Molly J. T. Kibenge, Marco Montes de Oca, Juan Pablo Pontigo, Yoandy Coca, Diego Caro, Karina Kusch, Rudy Suarez, Ian Burbulis, Frederick S. B. Kibenge

**Affiliations:** 1Laboratorio de Biotecnología Aplicada, Facultad de Ciencias Naturales, Escuela de Medicina Veterinaria, Sede de la Patagonia, Puerto Montt 5480000, Chile; 2Centro de Investigaciones Biológicas Aplicadas (CIBA), Puerto Montt 5480000, Chile; 3Department of Pathology and Microbiology, Atlantic Veterinary College, University of Prince Edward Island, 550 University Ave, Charlottetown, PEI C1A 4P3, Canada; 4Laboratorio de Energía Renovables y Residuos (LERR-UC), Escuela de Ingeniería, Pontificia Universidad Católica de Chile, Avenida Vicuña Mackenna 4860, Santiago 7820436, Chile; 5Programa de Magíster en Acuicultura, Facultad de Ciencias del Mar, Universidad Católica del Norte, Coquimbo 1281, Chile; 6Facultad de Medicina, Escuela de Medicina, Sede de la Patagonia, Puerto Montt 5480000, Chile

**Keywords:** infectious pancreatic necrosis virus (IPNV), Atlantic salmon, *Salmo salar*, quantitative trait locus (QTL), viral protein 2 (VP2) mutations

## Abstract

Infectious pancreatic necrosis (IPN), caused by IPNV, affects several species of farmed fish, particularly Atlantic salmon, and is responsible for significant economic losses in salmon aquaculture globally. Despite the introduction of genetically resistant farmed Atlantic salmon and vaccination strategies in the Chilean salmon industry since 2019, the number of IPN outbreaks has been increasing in farmed Atlantic salmon in the freshwater phase. This study examined gross and histopathological lesions of IPNV-affected fish, as well as the IPNV nucleotide sequence encoding the VP2 protein in clinical cases. The mortality reached 0.4% per day, and the cumulative mortality was from 0.4 to 3.5%. IPNV was isolated in the CHSE-214 cell line and was confirmed by RT-PCR, and VP2 sequence analysis. The analyzed viruses belong to IPNV genotype 5 and have 11 mutations in their VP2 protein. This is the first report of IPN outbreaks in farmed Atlantic salmon genetically resistant to IPNV in Chile. Similar outbreaks were previously reported in Scotland and Norway during 2018 and 2019, respectively. This study highlights the importance of maintaining a comprehensive surveillance program in conjunction with the use of farmed Atlantic salmon genetically resistant to IPNV.

## 1. Introduction

Infectious pancreatic necrosis virus (IPNV), an etiological agent of infectious pancreatic necrosis (IPN), is one of the important viral pathogens in salmonid aquaculture. IPNV belongs to the genus *Aquabirnavirus*, family *Birnaviridae*, Kingdom *Orthonavirae*, Real *Riboviria* [1]. IPNV is the prototype of the *Birnaviridae* family, which is characterized by nonenveloped particles of icosahedral symmetry measuring about 65 nm in diameter [2]. The IPNV genome consists of two linear double-stranded RNA (dsRNA) segments, designated A and B of approximately 3.0 and 2.9 kbp, respectively [3,4]. Segment A contains two open reading frames (ORFs), the larger one encoding a 104-kDa polyprotein (NH2-pVP2-VP4-VP3-COOH) that is cleaved into mature capsid proteins VP2 and VP3 and the viral protease VP4 [5]. The smaller ORF within segment A encodes VP5 [6], whereas segment B contains one ORF, which encodes VP1, the viral RNA-dependent RNA-polymerase [7].

Among the IPNV proteins, the structural protein VP2 is the main component of the viral capsid. VP2 organizes as trimers forming the spikes of the capsid, and therefore plays a crucial role in host cell recognition by the virus and infection [8]. Within the VP2 amino acid sequence, the range between positions 206 and 350 has been identified as a hypervariable region (HPR) [9] and contains neutralization epitopes [10]. Strains causing differing mortality from challenge experiments under controlled conditions have shown differences in their aminoacidic sequences, especially in positions within the HPR. These differences in VP2 protein sequence have been identified as virulence determinants [11,12]. Although amino acid positions 217, 221 and 247 stand out among others as having been identified in multiple studies [11,12,13,14,15,16], there is no consensus about the specific residues corresponding to either virulent or non-virulent IPNV strains. Interestingly, amino acid positions 217, 221 and 247 are located within the projection domain of the protein, possibly contributing to host cell recognition. Atlantic salmon (*S. salar*) resistance to IPN has a genetic factor [17], identified by quantitative trait locus (QTL) on chromosome 26 [18,19,20]. Mapping of this QTL by whole-genome sequencing has detected a region including an epithelial cadherin gene, whose protein product has been shown to bind with IPNV [21]. Thus, genetic variation at this epithelial cadherin gene explains, at least partially, the genetic factor seen in IPN resistance. The implementation of genetically resistant fish has led to a notable decrease in IPN cases. However, recent outbreaks of IPNV variants affecting genetically resistant fish have been reported in Scotland [22] and Norway [23]. Whether this recovery in IPNV’s capacity to infect genetically resistant fish is due to residue changes at the HPR of VP2 calls for further research. Because of this, identifying mutations in IPNV’s VP2 particularly located within the HPR is of major relevance as these could potentially explain the recent adaptation of the virus towards genetically resistant Atlantic salmon (*S. salar*).

IPNV was isolated for the first time in Chile in 1987 from Atlantic salmon (*Salmo salar*) fry from eggs imported from North America. Serological studies showed the virus related to the North American VR-299 serotype [24], later confirmed by nucleic acid and protein studies [25]. In 1995, the presence of IPNV was reported, mainly affecting Atlantic salmon smolt cultivated in Region X in the freshwater phase and fish farms after transfer to lakes and the sea. It was not until 1998 that the National Fisheries Service officially reported the situation [26]. Serological studies carried out by Fundación Chile in 1999 confirmed that the virus corresponded to the Sp serogroup, which was considered a new introduction, possibly from Europe, and confirmed by sequencing the 1998 isolate from Atlantic salmon [27] (Figure 1A). In 1999, injectable vaccines were introduced against IPNV, increasing the vaccinated population rapidly [28], which meant a significant decrease in cases after transfer to seawater, but the situation in freshwater farms persisted. It was not until the implementation of improvements in biosecurity and the introduction of genetically resistant Atlantic salmon (*S. salar*) to IPNV, identified by QTL on chromosome 26 [18,19,20], starting in 2010 that cases were reduced. As of 2019, an increase in IPN mortality in the freshwater phase has been observed in farmed Atlantic Salmon (Figure 1B).

New IPNV variants affecting genetically resistant farmed Atlantic salmon were first reported in Scotland [22] and then in Norway [23]. During the last decade, IPN cases have remained under control. The present study describes, for the first time, the epidemiology, pathology, and molecular virology of 17 clinical IPNV cases in genetically resistant Atlantic salmon farmed in Chile.

## 2. Materials and Methods

### 2.1. Sample Collection and Gross Pathology

Samples were collected from 11 IPN outbreaks recorded between 2020 and 2022. All cases occurred in Atlantic Salmon (*Salmo salar*) between the alevin and smolt stages grown in hatcheries located in Los Lagos and Magallanes regions using Recirculating Aquaculture System (RAS) technology (Table 1). Each hatchery maintained fish with genetic resistant strains (QTL-IPN) as part of a health strategy. Postmortem was performed on whole fish carcasses. The significant external and internal gross lesions were recorded and the frequency of each of these was determined as described by Noga [31].

### 2.2. Histopathology

Tissue samples for histological analysis were collected in 10% buffered formalin. They were then processed using standard procedures and the sections, 3–4 μm, were stained with hematoxylin and eosin (H&E) according to Prophet et al. [32] to describe the significant microscopic morphological changes.

### 2.3. Immunohistochemistry

Sections of the pyloric ceca and liver were fixed in 10% buffered formalin and subjected to immunohistochemistry staining to detect IPNV antigens. Antigen retrieval was performed with a steamer for 40 min, placing the slides in a 250 mL cuvette with EDTA recovery solution. Primary anti-IPNV OLIGOCLONAL antibody (ANGO, San Ramon, CA, USA) was used at a dilution of 1:500 μL. A secondary antibody conjugated with HRP (HiDef Detection™ polymer system, Cell Marque, Rocklin, CA, USA) was applied after a series of incubations and washes. Development was performed using 3,4-diaminobenzidine chromogen (ImmPACT^®^ DAB Substrate Kit, Vector Laboratories, Newark, CA, USA). The samples were observed by optical microscopy at 200× magnification using a Leica (Wetzlar, Germany) DM 2000 LED microscope.

### 2.4. RNA Extraction and RT-PCR

For the cases investigated, an automated tissue homogenization of samples was performed using the MagNA Lyser instrument (Roche, Basel, Switzerland). Total RNA was extracted using a robot (Roche MagNA Pure LC instrument) with the MagNA Pure LC RNA isolation kit III—Tissue (for the virus), according to the manufacturer’s instructions. The extracted RNA was eluted in 50 μL of nuclease-free water, RNA yields were quantified, and purity was analyzed using the OD260/280 ratio and a NanoPhotometer^®^ P 300 (Implen, Westlake Village, CA, USA). The eluted RNA was tested immediately following quantitation. The RT-qPCR analysis was done using Light-Cycler 480 RNA Master Hydrolysis Probes for RNA (Roche). For the detection of IPNV, thermocycling conditions and specific primers described by Ingerslev et al. [33] were used. For piscine orthoreovirus (PRV), the RT-qPCR was done using PRV-specific primers and conditions as described by Palacios et al. [34]. For the detection of infectious salmon anemia virus (ISAV) in tissue homogenates, ISAV-specific primers and conditions described by Snow et al. [35] were used.

### 2.5. IPNV Isolation and Cell Culture

Head kidney, spleen, and liver were sampled in fry and smolt and preserved in PBS containing Fetal Bovine Serum (FBS) 10% (Gibco, Grand Island, NY, USA) until cell inoculation. Tissues were inoculated on monolayers of Chinook salmon embryo (CHSE-214) cell line (ATCC CRL-1681) according to the protocol described previously [36]. CHSE-214 cells were cultured in Eagle’s minimum essential medium (EMEM) supplemented with 10% FBS, 1 × non-essential amino acids, and 2 µM L-glutamine at 20 °C with 4% CO_2_. Infected cells were examined microscopically daily until the emergence of cytopathic effect (CPE). IPNV presence was confirmed by RT-PCR according to Ingerslev et al. [33].

### 2.6. DNA Sequencing

Samples positive for IPNV by RT-qPCR with the lowest cycle threshold (Ct) values were utilized to perform DNA sequencing of the IPNV VP2 gene. RT-PCR was performed on IPNV segment A using the QIAGEN One-Step RT-PCR kit (Qiagen, Toronto, ON, Canada) under the following thermocycling conditions: reverse transcription at 50 °C for 30 min; denaturation at 95 °C for 15 min; 45 cycles of denaturation at 94 °C for 60 s, annealing at 54 °C for 60 s, and extension at 72 °C for 10 min; final extension at 72 °C for 10 min, and held at 4 °C indefinitely.

Primer IPNV-A1-F (5′-TGAGATCCATTATGCTTCCAGA-3′) and IPNV-A2-R (5′-CAGGATCATCTTGGCATAGT-3′) were used to amplify a 1179-bp fragment of the partial VP2 gene. Amplified PCR products were electrophoresed on 1.5% agarose gels to detect amplification of the VP2 fragment.

The PCR products were purified using a QIAquick Gel Extraction kit (Qiagen) and were then cloned into the pCRII vector using a TOPO TA cloning kit (Invitrogen, Life Technologies Inc., Burlington, ON, Canada) in preparation for DNA sequencing. Recombinant plasmid DNA for sequencing was prepared using the QIAprep Spin Miniprep kit (Qiagen). DNA was sequenced by Moxilabs (McMaster University, Hamilton, Ontario) using the Sanger sequencing method. The 1179-bp fragment of VP2 was assembled and blasted to the GenBank database using BLAST [37]. The IPNV sequences obtained were used for phylogenetic analysis. The partial VP2 gene sequences obtained in this study were deposited in the GenBank database under accession numbers provided in Appendix A.

### 2.7. Data Analysis

The percentage of mortality data due to IPN in Atlantic salmon cultivated in the freshwater phase from 2011 to 2021 was extracted from the National Fisheries Service [30]. IPNV-VP2 sequences of isolates from Chile were extracted from the GenBank database [29] and subjected to genotype assignment. The latter was performed by local alignments through EMBOSS water (v6.6.0.0) [38], assessing for the higher identity within query sequences and representative sequences of each IPNV genotype.

The hypervariable region (HPR) encompassing nucleotides 618 to 1050 were extracted from VP2 DNA sequences through EMBOSS water local alignments. Multiple sequence alignment (MSA) of HPR DNA and Protein sequences were aligned with Clustal Omega (v1.2.4) [39]. Phylogenetic analysis was performed under maximum likelihood with RAxML (v8.2.12) [40] and illustrated with iTOL (v6.5.8) [41]. The identification of mutations in IPNV VP2 sequences of isolates from the 2022 IPN outbreaks in Chile was performed through EMBOSS water local alignments of VP2 sequences compared with the 2012 IPNV VP2 sequence KU609583 from Chile. VP2 protein structure prediction was performed with AlphaFold2 [42] implemented in ColabFold (v1.3.0) [43]. VP2 mutations were mapped on the predicted structure with the Protein Imager interface (v0.5.60) [44].

## 3. Results

### 3.1. Gross Pathology

All fish sampled showed reduced feed intake, were lethargic, and showed superficial swimming with loss of equilibrium. Gross pathology was considered classical (Figure 2A), characterized by the presence of pale organs (gill, heart and liver), diffuse petechial hemorrhages (Figure 2B,C) in the pyloric caeca, and the absence of food in the gastrointestinal system. Less frequently, fish with mucous content in the stomach were observed. Mortality reported was variable, reaching 0.40% per day and the cumulative mortality was between 0.4–3.5%.

### 3.2. Histopathology

Table 2 shows the nature and frequency of significant histopathology lesions in tissues of Atlantic salmon affected by IPN between 2020 and 2022.

Among the most frequent findings are pancreatic necrosis, hepatic necrosis, and necrosis of the epithelium of the pyloric caeca and intestine (Figure 3A,B). Additionally, the virus is detected in the epithelium of the pyloric caeca and pancreas using immunohistochemistry (Figure 3C). Less frequently, nephrocalcinosis is observed in the kidney while myocarditis and epicarditis in the heart.

### 3.3. IPNV PCR

Positive qRT-PCR amplification was achieved in all tissue samples examined; these were used for subsequent infection of CHSE-214 cells. The primers used for viral amplification correspond to a partial VP2 gene fragment, from segment A, which were used for detection. Figure 4 shows the analysis of the average Ct values for all tissue samples by RT-PCR.

### 3.4. Cell Culture

The morphology of control CHSE-214 cells (Figure 5A) showed practically no variation during the evaluation period. On the contrary, the infected cells showed the appearance of round translucent cells with vacuoles from 12 h post infection (hpi), whose proportion increased to 48 hpi as seen in Figure 5B a 30% CPE versus a 90% CPE in cells infected with IPNV in Figure 5C. Microscopically, morphological changes in the cell monolayer are observed in infected cells with large numbers of round sloughed translucent cells evident in infected cells versus control cells. All infected cells confirmed infection by positive RT-qPCR with an average of Ct 25.

### 3.5. Phylogenetic Analysis

Phylogenetic analysis of IPNV VP2 nucleotide sequences showed that the isolates from the 2022 IPN outbreak in Chile clustered within IPNV isolates from Chile collected in 2010 and 2012 (Figure 6). All sequences from the 2022 IPN outbreak in Chile belong to genotype 5.

Sequences corresponding to IPNV isolates from Europe sampled between 2004 and 2019 formed a different cluster in genotype 5. Further, DNA sequence alignments between isolates from the 2022 IPN outbreaks and those from the 2010 and 2012 IPN outbreaks in Chile showed higher identity percentages than alignments between isolates from the 2022 IPN outbreaks in Chile and the IPNV isolates from Europe (Figure 7), suggesting that the isolates from the 2022 IPN outbreaks in Chile are variants originating from local IPNV strains rather than variants from the recent IPNV strains from Europe.

### 3.6. VP2 Mutations

The analysis of IPNV VP2 amino acid sequences from the 2022 IPN outbreaks in Chile showed 11 amino acid changes, with most of them within the hypervariable region (Table 3). When compared with the 2012 IPNV VP2 sequence (KU609583) from Chile, the mutations K12R, T199I, A217T, T221A, T247A, and D282N were found with a frequency between 60% and 100%. However, the mutations G61D, G96A, P209S, L213P, and E379G were found with a frequency of 10%. On the other hand, when compared with the 2014 IPNV VP2 sequence (OP585397) from Chile, the mutations T221A, and T247A were found with frequencies of 80% and 100%, respectively, whereas mutations G61D, G96A, P209S, L213P, T217A, N282D, and E379G were found with a frequency between 10% and 20%.

To further explore the potential effect of the identified mutations, we performed a structural prediction of the VP2 protein from the OP329390 2022 isolate. The mutations from all IPNV isolates from the 2022 IPN outbreaks in Chile were mapped within the obtained structure (Figure 8). Mutations T199I, P209S, L213P, A217T, T221A, T247A and D282N fall within the projection domain, involved in host cell recognition whereas mutations K12R, G61D and G96A are located within the shell domain, which constitutes the main layer of the IPNV capsid. Mutations G61D, G96A and L213P fall within positions mostly buried in the VP2 structure but as their residue characteristics notably change, these mutations could alter the protein structure locally. In contrast, mutations K12R, and D282N might maintain the protein local structure due to their similar residue features. Nevertheless, mutations T199I, A217T, T221A and T247A shift between polar and hydrophobic side chains, potentially modulating binding properties as all residue positions are exposed.

## 4. Discussion

In Chile, IPNV has been reported, belonging to genotype 1, mainly affecting the farming of rainbow trout (*O. mykiss*) and genotype 5, mainly affecting the farming of Atlantic Salmon (*S. salar*) [24,25,27]. After 1995, a predominance of the frequency of genotype 5 was observed, possibly associated with the virulence, susceptibility of the species and the gradual predominance of Atlantic Salmon farming. After the implementation of the use of vaccines, implementation of biosecurity measures, and the use of genetically resistant eggs, a significant reduction in mortality was observed in the culture phase in freshwater and seawater, reaching a minimum of losses in the phase cultured in fresh water in 2018. However, as of 2019, a gradual increase in infectious losses due to IPNV in Atlantic Salmon has been observed, reaching a mortality rate of 17.1% in 2021 due to infectious causes [30]. As of 2020, cases of IPN have been reported in genetically resistant batches, QTL-IPNV. The affected batches have presented accumulated mortalities between 0.8% and 8.1% but still significantly lower than those reported prior to the implementation of prevention and control measures. Additionally, it should be considered that the mortality reported officially [30] includes genetically resistant and non-resistant batches. The clinical signs, macroscopic pathology, and histopathology of the cases analyzed are similar to those classically described for the disease [14,45,46,47,48,49].

Viral load measurements from tissue samples are indicative of active virus replication and are routinely used to monitor severe viral respiratory tract infections, including clinical progression, response to treatment, cure, and relapse [50]. The average Ct of the cases analyzed corresponded to 23.7, with a standard deviation of 5.09. The dispersion of the Ct observed in the clinical samples corresponding to the cases analyzed in this study can be explained by the fact that the samples were extracted even when the selected fish were in different stages of the course of the disease.

The clinical samples of tissue from selected fish inoculated in CHSE-214 presented the classic cellular morphological changes of IPN infection, the presence of the IPNV was confirmed by RT-PCR [36].

In the present study, the sequence analysis of IPNV isolates from the 2022 IPN outbreaks in Chile showed them to be more similar to IPNV isolates from Chile in 2010, 2012, and 2014 than the recent IPNV isolates from Europe (Figure 7), suggesting that the isolates from the 2022 IPN outbreaks are variants originating from local IPNV strains rather than from recent IPNV strains from Europe.

Our study further indicates that the IPNV isolates from the 2022 outbreaks in Chile have evolved at the projection domain of VP2, which is likely involved in host cell recognition [8]. Both vaccination and genetically resistant fish using QTL strategies have been implemented in Chile, and our analysis suggests that 2022 IPNV isolates in Chile have evolved to adapt against the selective pressure these two programs represent. Other risk factors such as biosecurity and management strategies should also be considered.

Amino acid positions 217, 221 and 247 of the VP2 protein had been previously reported as associated with virulent and non-virulent strains of IPNV mainly in Atlantic salmon [11,12,13,14,15,16]. In position 217 we found the residue T in nine out of ten sequences described in this study. Threonine 217 has been related with high virulent strains [11,12,13,15,51]. Further, in position 221, we identified the residue A in eight out of ten sequences reported here. Alanine 221 has been described as part of highly virulent strains as well [11,13,14,15]. Additionally, position 247 corresponds to the residue A in all ten sequences analyzed here. Alanine 247 has been related to low virulent strains [12,15], but also with high virulent strains [52]. Similarly, in position 199 all ten sequences from this study show the residue I. Isoleucine 199 has been associated with low virulent strains [12], but also with high virulent strains [14]. Moreover, when compared with the 2012 IPNV VP2 sequence (KU609583) from Chile, sequences from 2022 show amino acid changes at positions 12, 61, 96, 209, 213, 282 and 379. Although these changes were found mostly with low frequency, to the best of our knowledge these have not been previously related with high virulence.

Similar situations have been described with isolated IPNV from 2018 in Scotland [22] and isolated IPNV from 2019 in Norway [23]. Even though these strains have been identified as persistent and low virulent based on their VP2 amino acid virulent determinants, these have shown the ability to generate IPN outbreaks in QTL-resistant fish.

Altogether, these observations suggest that the amino acid patterns present in VP2 are relevant for virulence to the extent that the obtained structure is functionally advantageous, in relation to its ability to bind to the host and/or promote immune evasion.

These studies show how fortuitous and spontaneous the emergence of new variants is and the wide spectrum of mutations in the VP2 protein that can be related to IPNV virulence. To elucidate whether the identified mutations have a direct impact on the host receptor interaction with VP2 and the pathogenicity of IPNV requires further research.

## Figures and Tables

**Figure 1 pathogens-11-01368-f001:**
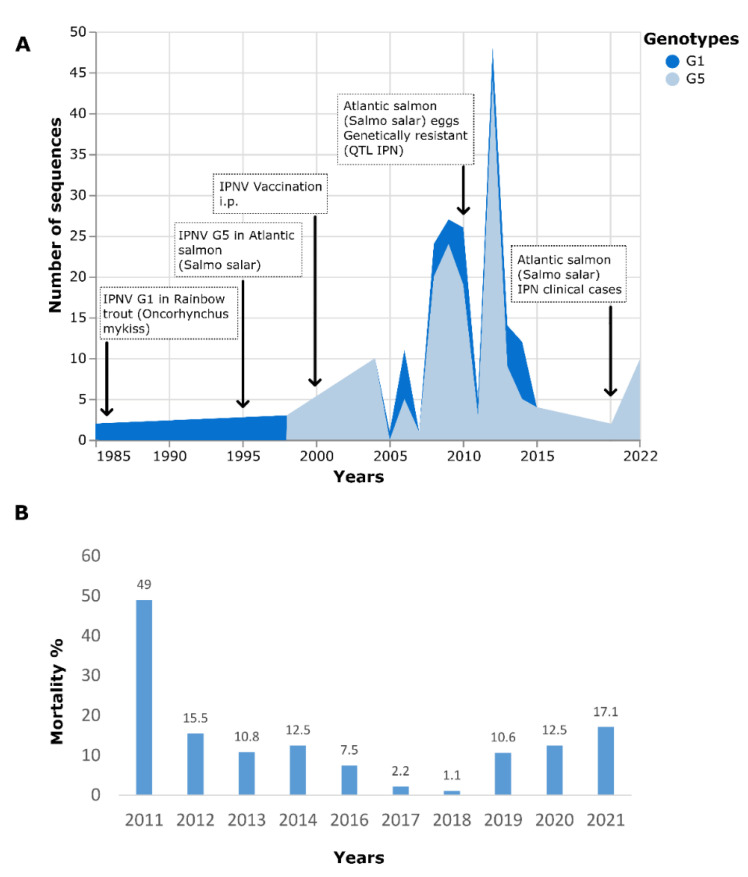
(**A**) Evolution of genotype frequency of IPNV sequences available in the GenBank database [29] and the main milestones regarding the IPN situation in Chile; (**B**) Percentage of mortality due to IPN in Atlantic salmon cultivated in the freshwater phase (2011–2021) (Source: National Fisheries Service, 2022 [30]).

**Figure 2 pathogens-11-01368-f002:**
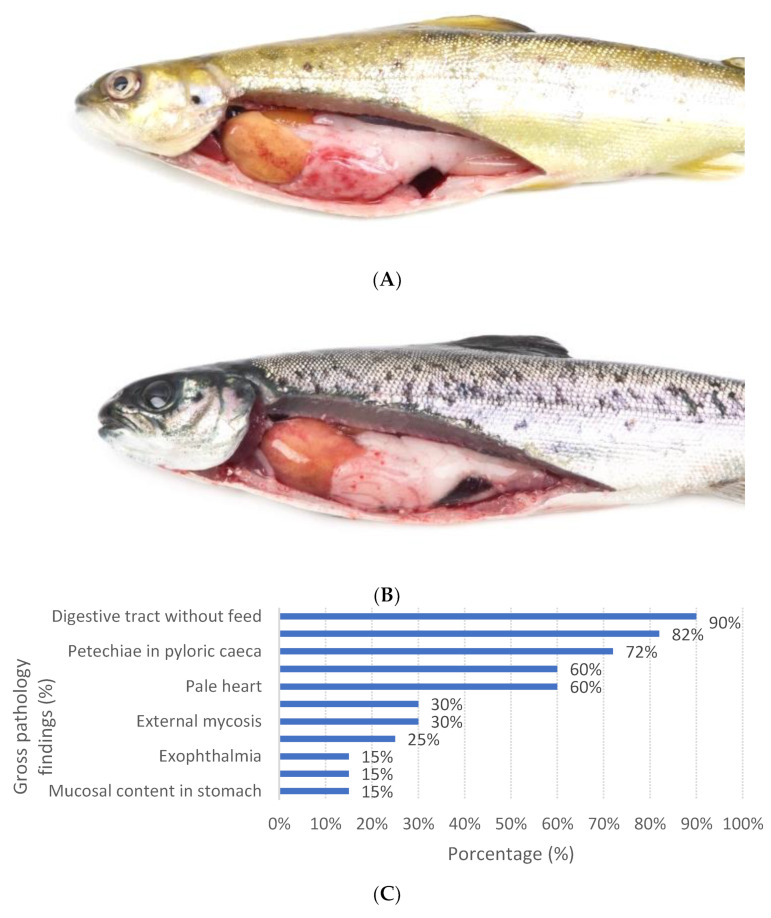
(**A**) Smolt from the IPN outbreak in 2022 showed diffuse hemorrhages in visceral fat and yellow liver. (**B**) Smolt from the IPN outbreak in 2022 showing petechial hemorrhages in pyloric fat and yellow liver. (**C**) Frequency of gross pathology findings in Atlantic salmon (Salmon salar) diagnosed with IPN from 2020 to 2022.

**Figure 3 pathogens-11-01368-f003:**
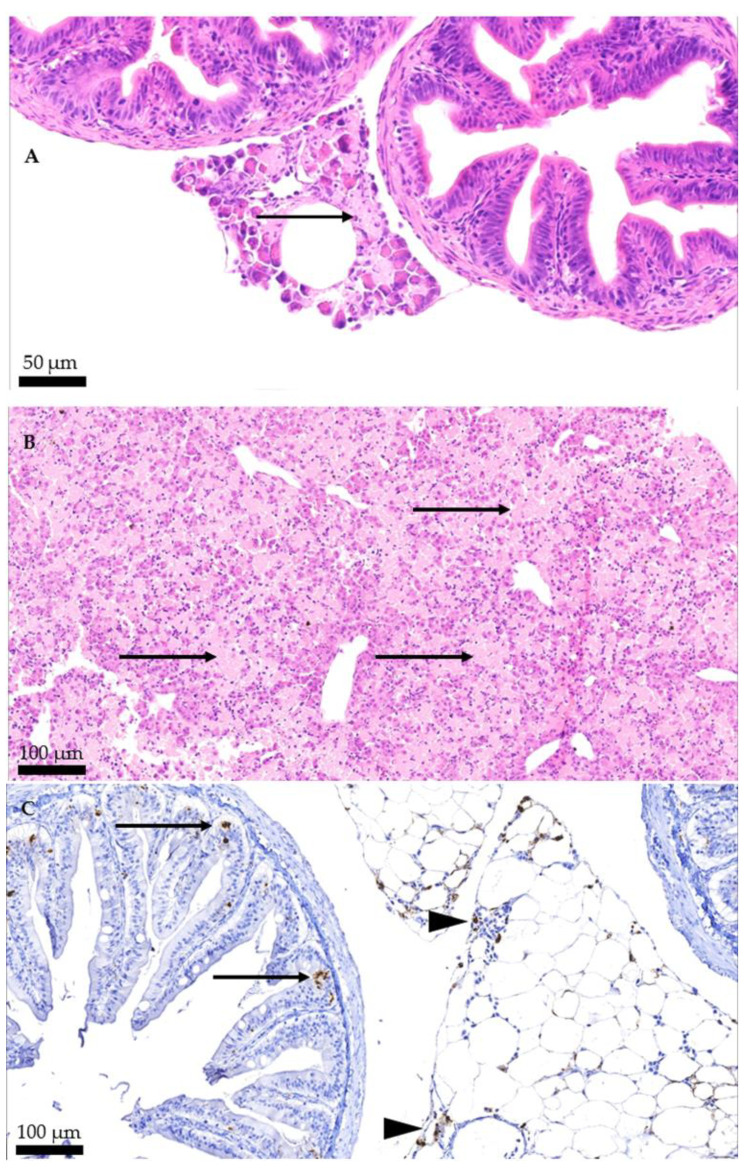
Atlantic salmon (*Salmo salar*) affected by IPN. (**A**) Pyloric caeca and pancreas (H&E). The arrow indicates diffuse pancreatic necrosis, characterized by degeneration of acinar cells, and formation of amorphous eosinophilic material, necrosis. (**B**) Liver (H&E). The arrows indicate diffuse, severe, hepatic necrosis. (**C**) Immunohistochemical positive staining of the pyloric caeca and pancreas, using IPNV-specific antibodies, brown show the IPNV presence. Cells infected with IPNV are located mainly in the pancreatic tissue (arrowheads), but also in some epithelial cells in the pyloric ceca (arrows).

**Figure 4 pathogens-11-01368-f004:**
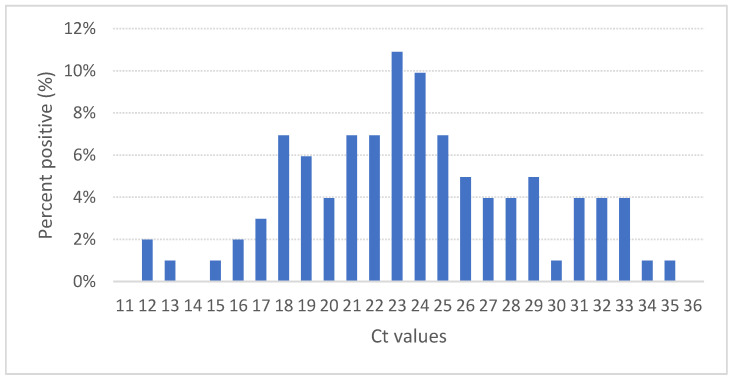
Frequency distribution of the cycle threshold (Ct) values of the clinical cases included in this study.

**Figure 5 pathogens-11-01368-f005:**
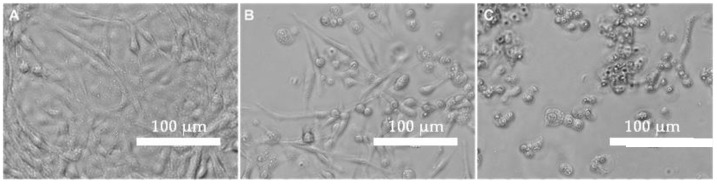
Cytopathic effects on the CHSE-214 cell line infected with IPNV isolates. (**A**): Control cell grown under optimal conditions at 48 hr. (**B**): Low cytopathic effect at 12 hpi. (**C**): High cytopathic effect at 48 hpi.

**Figure 6 pathogens-11-01368-f006:**
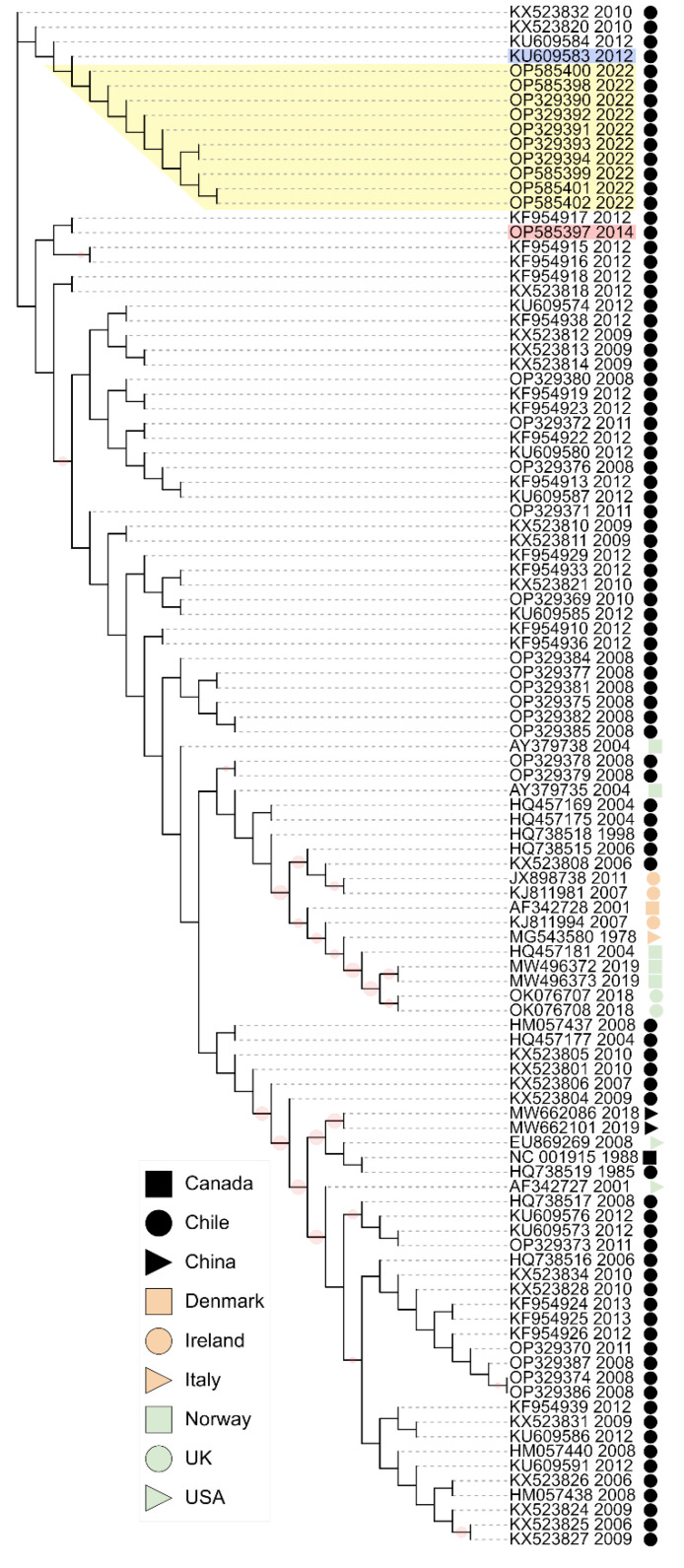
Phylogenetic tree of the hypervariable region from VP2 nucleotide sequences. The HPR encompassing nucleotides 618 to 1050 was extracted from VP2 DNA sequences. HPR sequences were compared through multiple sequence alignment. A phylogenetic analysis was performed under maximum likelihood. IPNV sequences from the 2022 IPN outbreaks in Chile are highlighted in yellow. Sequences from 2012 and 2014 later used to contrast aminoacidic mutations are show in blue and red, respectively. Bootstrap *n* = 1000. Frequencies ≥70 are shown as red circles. GenBank accession numbers and collection years are also shown.

**Figure 7 pathogens-11-01368-f007:**
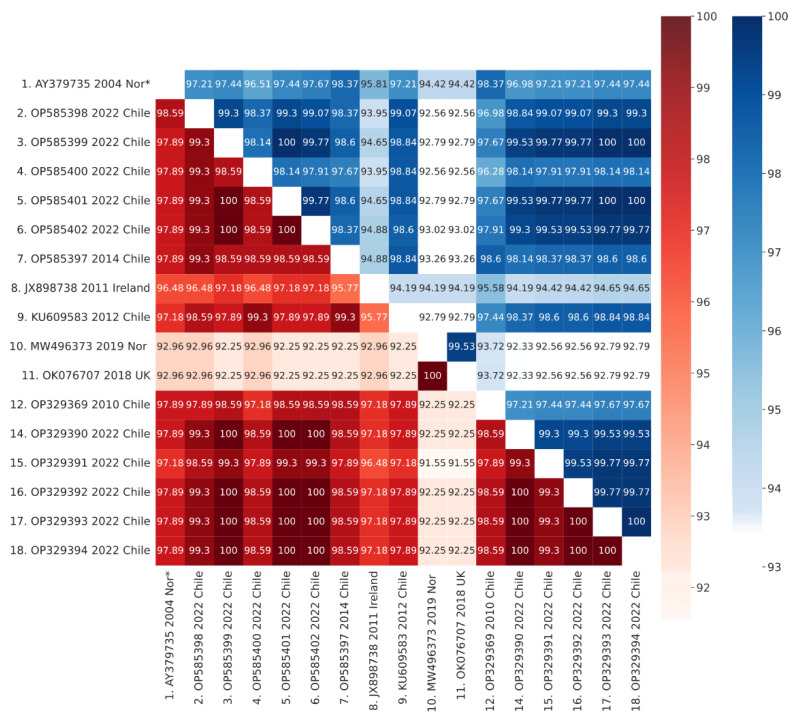
DNA sequence identities (blue scale) and protein sequence similarities (red scale) of the hypervariable region from recent Chilean outbreaks and representative isolates from Europe. Nor* denotes Norway.

**Figure 8 pathogens-11-01368-f008:**
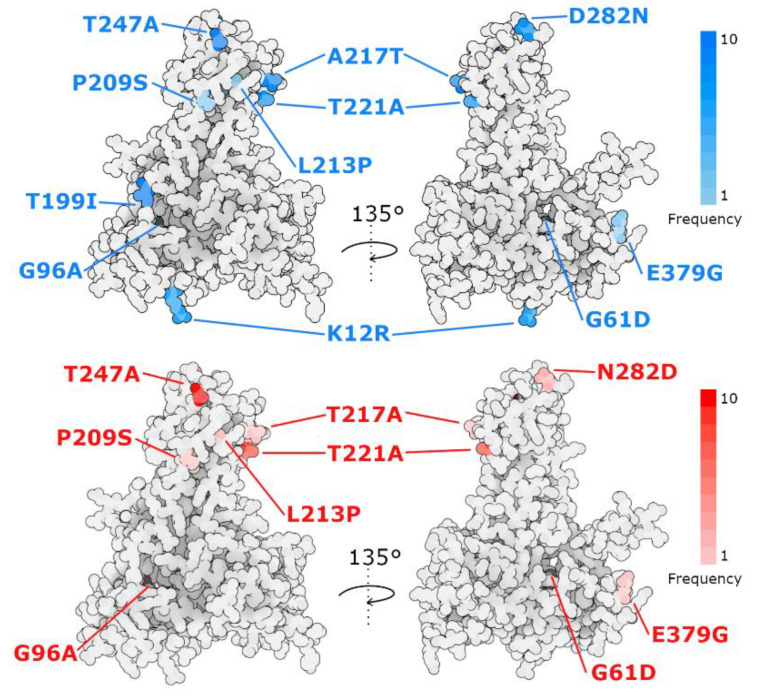
VP2 Amino acid substitutions identified from 2022′s IPNV outbreaks in their structural context. Structure predicted from VP2 amino acid sequence of IPNV isolate OP329390 using ColabFold. **Top**: mutations found in sequences from 2022 when compared with the Chilean VP2 sequence from 2012 (KU609583). **Bottom**: mutations observed in sequences from 2022 when compared with the Chilean VP2 sequence from 2014 (OP585397).

**Table 1 pathogens-11-01368-t001:** General epidemiological background and laboratory results of the cases analyzed in this study.

Case ID	3987	3759	4258	4257	4579	5117	5126	7023	7036	7027	7061	3746	7150	7190	7189	7176	95
Date	01-2020	09-2020	05-2021	05-2021	10-2021	05-2022	05-2022	06-2022	06-2022	06-2022	06-2022	08-2020	08-2022	08-2022	08-2022	08-2022	2014
Region	IX	X	X	IX	X	XI	X	X	XI	X	X	X	X	X	X	X	X
Salinity	FW	BW	BW	FW	BW	FW	BW	BW	FW	BW	BW	FW	FW	FW	BW	FW	FW
Development stage	Fry	Pre Smolt	Pre Smolt	Fry	Pre Smolt	Fry	Pre Smolt	Pre Smolt	Fry	Pre Smolt	Pre Smolt	Fry	Pre Smolt	Pre Smolt	Pre Smolt	Pre Smolt	NA
Mortality (%)	8.1	0.83	1.2	5	0.95	9.7	0.7	0.45	4.3	1.15	0.78	3.1	0.8	0.93	2.7	1.2	NA
Outbreak time (weeks)	4 to 5	3	3 to 4	4 to 5	3 to 4	4 to 5	3	3	4	3 to 4	3	3 to 4	2 to 3	3 to 4	3 to 4	3 to 4	NA
Fish sampled	8	20	10	10	13	3	5	17	15	8	6	6	2	19	13	10	NA
Average IPNV Ct	20.5	28.1	23.2	25.4	25.6	13.3	27.4	25.4	21.2	21.5	17.4	30.5	26.5	20.2	20.2	27.6	21.9
Cell culture	CPE	NA	CPE	NA	NA	NA	NA	CPE	NA	NA	NA	CPE	NA	NA	CPE	NA	NA
GenBank accession	NA	NA	NA	NA	NA	OP329390	NA	NA	NA	OP585398	OP329391 OP329392 OP329393 OP329394	NA	NA	OP585401OP585402	OP585399OP585400	NA	OP585397

NA denotes does not apply.

**Table 2 pathogens-11-01368-t002:** Nature and frequency of significant histopathology lesions in tissues of Atlantic salmon affected with IPN between 2020 and 2022.

Tissue	Histopathological Description	Diagnosis	Frequency
Pancreas	Structural loss of the acinar cells of the pancreas, forming amorphous eosinophilic mass areas.	Diffuse, severe, pancreas necrosis.	37/46
Liver	Structural loss of multiple coalescent areas of hepatocytes, in some cases, it presents as ballooning degeneration, characterized by enlarged and swollen hepatocytes with granular material in the cytoplasm. In some, a loss of cell integrity is observed, and pyknotic nuclei and eosinophilic bodies are consistent with apoptosis.	Diffuse, severe, hepatic necrosis.	36/46
Pyloric ceca/intestine	Necrosis and sloughing of epithelium. Eosinophilic content in the lumen of the pyloric ceca.	Diffuse, mild to severe, intestinal necrosis. NA	5/46 6/46
Peripiloric fat	Multiple hemorrhagic foci, fat cell boundary loss and loss of their peripheral nuclei. In some cases, their cytoplasm has become a pink amorphous mass of necrotic material.	Moderate to severe focal to multifocal hemorrhagic fat necrosis.	3/46
Kidney	Presence of basophilic amorphous deposits in lumen of tubules and ureters.	Renal calcinosis, mild to moderate.	3/46
Heart	Multifocal to diffuse infiltration of mononuclear cells of myocardium and epicardium.	Multifocal to diffuse, subacute, mild to moderate, epicarditis and myocarditis.	8/46

NA denotes does not apply.

**Table 3 pathogens-11-01368-t003:** Amino acid changes identified in VP2 protein.

	12	61	96	199	209	213	217	221	247	282	379
AY379735 2004 Norway	K	G	G	I	P	L	T	T	A	N	E
OP585398 2022 Chile	R	G	A	I	S	L	T	T	A	D	E
OP585399 2022 Chile	R	G	G	I	P	L	T	A	A	N	G
OP585400 2022 Chile	R	G	G	I	P	L	A	T	A	D	E
OP585401 2022 Chile	R	G	G	I	P	L	T	A	A	N	E
OP585402 2022 Chile	R	G	G	I	P	L	T	A	A	N	E
OP585397 2014 Chile	R	G	G	I	P	L	T	T	T	N	E
JX880108 1999 Ireland	-	S	G	I	P	L	P	T	P	A	E
KU609583 2012 Chile	K	G	G	T	P	L	A	T	T	D	E
MW496373 2019 Norway	K	G	G	I	P	L	P	T	A	T	E
OK076707 2018 UK	K	G	G	I	P	L	P	T	A	T	E
OP329369 2010 Chile	R	G	G	I	P	L	T	A	T	N	E
OP329387 2008 Chile	R	G	G	I	P	L	A	T	A	A	D
OP329390 2022 Chile	R	G	G	I	P	L	T	A	A	N	E
OP329391 2022 Chile	-	G	G	I	P	P	T	A	A	N	E
OP329392 2022 Chile	-	G	G	I	P	L	T	A	A	N	E
OP329393 2022 Chile	-	D	G	I	P	L	T	A	A	N	E
OP329394 2022 Chile	-	G	G	I	P	L	T	A	A	N	E

## Data Availability

GenBank accession numbers of partial VP2 gene sequences can be found at Appendix A.

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
