# Peer review of "Isolation of a New Infectious Pancreatic Necrosis Virus (IPNV) Variant from Genetically Resistant Farmed Atlantic Salmon (Salmo salar) during 2021–2022"

_pathogens, 2022, doi:10.3390/pathogens11111368_

Round 1
Reviewer 1 Report
In this study, their authors identified new IPNV from genetically resistant farmed Atlantic salmon and studied the reason why IPN outbreak happened through identification of mutated VP2 peptide. This manuscript is constructed well and give new insights for understanding IPN outbreak during 2021-2022.
Minor point
1. Their authors has to reformat all of tables by the guideline of table preparation
2. In Figure 2, the position of labeling as A, B and C was right-side. This position is weird. And in the C, labelling for description regarding to the graph should make them more clearer.
3. The scale bar in Figure 3 are too small to see.
4. In legend of Figure 3, their authors described right and left arrow. However, in this figure, right arrows were only.
5. In Figure 5, the unit of scale bar was not described.
6. The meaning of Ct value is missing in this manuscript.
7. In the legend of Figure 6, the bracket outside "HRP" should be removed.
8. Their authors identified mutated VP2. In vitro data regarding to the virulence of mutated VP2 will support to describe the significance of new INPV if their author have any in vitro data.
Author Response
General comment
In this study, their authors identified new IPNV from genetically resistant farmed Atlantic salmon and studied the reason why IPN outbreak happened through identification of mutated VP2 peptide. This manuscript is constructed well and give new insights for understanding IPN outbreak during 2021-2022.
Response: We appreciate your comment.
Minor point 1
Their authors has to reformat all of tables by the guideline of table preparation.
Response: Thank you for your reminder. We have edited Table 2 and now it is horizontally oriented. Table 3 was inserted into the main text.
Minor point 2
In Figure 2, the position of labeling as A, B and C was right-side. This position is weird. And in the C, labelling for description regarding to the graph should make them more clearer.
Response: Thank you for your remark. Description of part C has been corrected (line 242).
Minor point 3
The scale bar in Figure 3 are too small to see.
Response: Thank you for your comments. The total length of scale bars is now indicated in a bigger font size.
Minor point 4
In legend of Figure 3, their authors described right and left arrow. However, in this figure, right arrows were only.
Response: Thank you for your remark. Arrows and now also arrowheads are being used to better point different locations in part C.
Minor point 5
In Figure 5, the unit of scale bar was not described.
Response: Thank you for your reminder. The total length of scale bars is now indicated in a bigger font size.
Minor point 6
The meaning of Ct value is missing in this manuscript.
Response: Thank you for your remark. The meaning of Ct is now indicated at line 177.
Minor point 7
In the legend of Figure 6, the bracket outside "HRP" should be removed.
Response: Thank you for your reminder. Brackets outside of HPR were removed from this legend. Also, we noticed that the sample name OP329369 was mistakenly indicated as collected from the year 2012. This was corrected to the year 2010 at Figure 6.
Minor point 8
Their authors identified mutated VP2. In vitro data regarding to the virulence of mutated VP2 will support to describe the significance of new INPV if their author have any in vitro data.
Response: Thank you very much for your comment. We are planning experiments to compare the virulence of recent Chilean IPNV strains with strains from previous years. These experiments are projected to be part of a subsequent article, and therefore beyond the scope of this study. Also, data regarding the virulence of mutated VP2 at the HPR has been already shown previously by Santi et al., 2004 and Shivappa et al., 2004, suggesting that amino acid changes at these positions are likely related with virulence.
Reviewer 2 Report
In this manuscript, the high lethality of IPNV, the identification method used in this article, and the future significance of the first report of an IPN outbreak in farmed Atlantic salmon genetically resistant to IPNV are described in a comprehensive summary.
There are no major issues with the whole article, and few comments below need to be confirmed:
In Introduction:
Line49-57: The second paragraph of the introduction mainly describes the important role of structural protein VP2 in the recognition of host cells and infection by the virus in IPNV proteins. In subsequent experiments, PCR amplification based on part of the a segment of VP2 gene is mentioned, which can be emphasized in the introduction for its significance rather than function
Line81-84: As my suggesttion, the introduction should add the innovation of this study. Although it is the first study on IPN outbreak in cultured Atlantic salmon resistant to IPNV, it should also add what are the differences in our research methods, why we use such methods and how to prove the significance of this study
In Materials and Methods:
Line131: The presence of IPNV has been confirmed by PCR, but minor modifications are mentioned, which should be explained
In Results:
Line245: Referring to Table 3, which shows the DNA sequence identity and protein sequence similarity of the high variation region (HPR) of the recent Chilean outbreak and 249 representative isolates from Europe, and also shows that 245 isolates from the 2022 IPN outbreak in Chile were from 246 local IPNV strains, I think it is possible to add a table of protein sequence and nucleic acid sequence similarities of isolates from the 2022 IPN outbreak and the 2010 and 2012 IPN outbreak isolates from 243 Chile.
Author Response
General comment
In this manuscript, the high lethality of IPNV, the identification method used in this article, and the future significance of the first report of an IPN outbreak in farmed Atlantic salmon genetically resistant to IPNV are described in a comprehensive summary.
Response: We appreciate your comment.
Minor point 1
Line49-57: The second paragraph of the introduction mainly describes the important role of structural protein VP2 in the recognition of host cells and infection by the virus in IPNV proteins. In subsequent experiments, PCR amplification based on part of the a segment of VP2 gene is mentioned, which can be emphasized in the introduction for its significance rather than function.
Response: Thank you very much for your comment. The introduction is now including more background emphasizing the relevance of the HPR (lines 55-75).
Minor point 2
Line81-84: As my suggesttion, the introduction should add the innovation of this study. Although it is the first study on IPN outbreak in cultured Atlantic salmon resistant to IPNV, it should also add what are the differences in our research methods, why we use such methods and how to prove the significance of this study.
Response: Thank you for your remark. The introduction is now highlighting the relevance of this study regarding the taken multidisciplinary approach (lines 118-121).
Minor point 3
Line131: The presence of IPNV has been confirmed by PCR, but minor modifications are mentioned, which should be explained.
Response: Thank you very much for your comment. Previously, we mistakenly cited a paper for traditional RT-PCR. Now we have updated the reference used for RT-qPCR. Also, in line 316 we have corrected the results descriptions. It was previously mentioned that the same pair of primers were used for IPNV detection and Sanger sequencing. This is not the case and was corrected.
Minor point 4
Line245: Referring to Table 3, which shows the DNA sequence identity and protein sequence similarity of the high variation region (HPR) of the recent Chilean outbreak and 249 representative isolates from Europe, and also shows that 245 isolates from the 2022 IPN outbreak in Chile were from 246 local IPNV strains, I think it is possible to add a table of protein sequence and nucleic acid sequence similarities of isolates from the 2022 IPN outbreak and the 2010 and 2012 IPN outbreak isolates from 243 Chile.
Response: Thank you very much for your remark. Both DNA and protein sequences of Chilean samples from 2010 and 2012 were present in Table 3. However, to improve the readability of this results, we have changed Table 3 into a heatmap corresponding to Figure 7. In this heatmap, it is clearer which values correspond to either DNA identity or protein similarity values, as these are represented with blue or red colors respectively. Also, we have removed the sample OP329387 2008 Chile from Figure 7 (previously Table 3) as it corresponds to IPNV genotype 1, and the study focused on IPNV genotype 5.